# Transcriptional Activation of *Chac1* and Other Atf4-Target Genes Induced by Extracellular l-Serine Depletion is negated with Glycine Consumption in Hepa1-6 Hepatocarcinoma Cells

**DOI:** 10.3390/nu12103018

**Published:** 2020-10-02

**Authors:** Momoko Hamano, Shozo Tomonaga, Yusuke Osaki, Hiroaki Oda, Hisanori Kato, Shigeki Furuya

**Affiliations:** 1Department of Bioscience and Bioinformatics, Faculty of Computer Science and Systems Engineering, Kyushu Institute of Technology, 680-4 Kawazu, Iizuka, Fukuoka 820-8502, Japan; 2Laboratory of Functional Genomics and Metabolism, Graduate School of Bioresource and Bioenvironmental Sciences, Kyushu University, Fukuoka 819-0395, Japan; osaki.yusuke.315@s.kyushu-u.ac.jp; 3Division of Applied Biosciences, Graduate School of Agriculture, Kyoto University, Kyoto 606-8502, Japan; tomonaga.shozo.4n@kyoto-u.ac.jp; 4Laboratory of Nutritional Biochemistry, Graduate School of Bioagricultural Sciences, Nagoya University, Nagoya 464-8601, Japan; hirooda@agr.nagoya-u.ac.jp; 5Health Nutrition Laboratory, Department of Applied Biological Chemistry, Graduate School of Agricultural and Life Sciences, The University of Tokyo, Tokyo 113-8657, Japan; akatoq@mail.ecc.u-tokyo.ac.jp; 6Innovative Bio-Architecture Center, Faculty of Agriculture, Kyushu University, Fukuoka 819-0395, Japan

**Keywords:** l-serine deficiency, *Chac1*, Phgdh, Shmt, hepatoma

## Abstract

Mouse embryonic fibroblasts lacking D-3-phosphoglycerate dehydrogenase (Phgdh), which catalyzes the first step of de novo synthesis of l-serine, are particularly sensitive to depletion of extracellular L-serine. In these cells, depletion of l-serine leads to a rapid reduction of intracellular L-serine, cell growth arrest, and altered expression of a wide variety of genes. However, it remains unclear whether reduced availability of extracellular l-serine elicits such responses in other cell types expressing *Phgdh*. Here, we show in the mouse hepatoma cell line Hepa1-6 that extracellular l-serine depletion transiently induced transcriptional activation of Atf4-target genes, including cation transport regulator-like protein 1 (*Chac1)*. Expression levels of these genes returned to normal 24 h after l-serine depletion, and were suppressed by the addition of l-serine or glycine in the medium. Extracellular l-serine depletion caused a reduction of extracellular and intracellular glycine levels but maintained intracellular l-serine levels in the cells. Further, Phgdh and serine hydroxymethyltransferase 2 (Shmt2) were upregulated after l-serine depletion. These results led us to conclude that the Atf4-mediated gene expression program is activated by extracellular l-serine depletion in Hepa1-6 cells expressing *Phgdh*, but is antagonized by the subsequent upregulation of l-serine synthesis, mainly from autonomous glycine consumption.

## 1. Introduction

L-serine (Ser), a nutritionally dispensable amino acid, serves as an indispensable metabolic precursor for various biomolecules essential to fundamental cellular processes. These molecules include proteins, other amino acids, tetrahydroxy folate derivatives, membrane lipids, and nucleotides. Ser is synthesized *de novo* from 3-phosphoglycerate via a pathway called the phosphorylated pathway, which is catalyzed by 3-phosphoglycerate dehydrogenase (Phgdh), phosphoserine aminotransferase 1 (Psat1) and phosphoserine phosphatase (Psph). Ser is reversibly converted to glycine (Gly) catalyzed by serine hydroxymethyltransferase (Shmt) 1 and 2. Recent studies demonstrated that in cancer cells, Phgdh and Shmt1/2 play key roles in regulating proliferation via synthesis of Ser and Gly. This is due to their roles as precursors of purine nucleotides and ATP production that contribute to folic acid metabolism, and glutathione (GSH) which regulates intracellular redox homeostasis [1,2,3]. In addition to these well-established roles in the cancer metabolism, ample evidence highlights that Ser and Gly metabolism contributes to epigenetic modifications of DNA and RNA via maintaining mitochondrial one carbon pathway and SAM in cancer cells [4,5].

We previously demonstrated that extracellular Ser deficiency led to cell growth arrest and cell death via phosphorylation of p38 MAPK, which was activated by an accumulation of 1-deoxy-sphinganine (doxSA) to reduce Ser availability in *Phgdh*-deficient mouse embryonic fibroblast (KO-MEFs) [6,7]. Moreover, Ser depletion drove the integrated stress response (ISR) pathway and altered gene expression profiles in KO-MEFs [8], and enhanced vulnerability to oxidative stress resulting in inflammation [9]. Our microarray results demonstrated that intracellular Ser deficiency induced the transcriptional activation of Atf4-target genes including cation transport regulator-like protein 1 (*Chac1*) in KO-MEFs [8]. These studies shed light on the cellular responses resulting from Ser depletion in non-malignant normal cells. However, whether reduced availability of extracellular Ser elicits such alterations of gene expression in *Phgdh*-expressing cells remains unknown.

To clarify response mechanisms to external Ser depletion in cells expressing *Phgdh*, we examined gene expression, intracellular Ser level, and cell proliferation in a mouse hepatoma Hepa1-6 line expressing *Phgdh*. Here, we demonstrate that extracellular Ser depletion causes transient transcriptional activation of Atf4-target genes *Chac1*, which associates with subsequent recovery of intracellular Ser levels and Gly consumption by upregulation of Phgdh and Shmt2.

## 2. Materials and Methods

### 2.1. Cell Culture and Treatment Condition

Mouse Hepa1-6 hepatoma cells (hepatoma cell 78 line, ACTT^®^ CRL-1830) were cultured in Dulbecco’s Modified Eagle Medium (DMEM; Wako Pure Chemical Industries Ltd., Osaka, Japan) containing 10% fetal bovine serum (FBS; Gibco, Thermo Fisher Scientific, Waltham, MA, USA), 4 mM glutamine, and 10 μg/mL gentamicin (Nacalai Tesque, Kyoto, Japan) at 37 °C and 5% CO_2_ as previously described [10].

For Ser depletion, Hepa1-6 were placed in Eagle’s Minimal Essential Medium (EMEM; Wako Pure Chemical Industries Ltd. Osaka, Japan) containing 1% FBS as previously described [10]. EMEM contained all essential amino acids as well as L-Gln but did not include L-Ala, L-Asp, L-Asn, L-Cys, L-Glu, Gly, L-Pro and L-Ser. This medium contains 4 μM Ser, which is derived from 1% FBS. Additives of Ser and Gly were prepared as 200 mM stock solutions and were added in EMEM to a final concentration of 400 μM.

### 2.2. Total RNA Extraction and Quantitative Analysis of mRNA Expression

Total RNA was extracted from Hepa1-6 with the High Pure RNA Isolation Kit (Roche, Basel, Switzerland) according to the manufacturer’s instructions. Following isolation, 1 μg of DNase-treated RNA was used to generate cDNA by reverse transcription using the High Capacity cDNA Reverse Transcription kit (Applied Biosystems, Life Technologies Japan Ltd., Tokyo, Japan). Quantitative real-time PCR was performed with a Model M×3000P Real Time PCR system (Agilent Technologies Japan Ltd., Tokyo, Japan) containing Thunderbird SYBR qPCR Mix (TOYOBO, Osaka, Japan) and the reference dye ROX according to the manufacturer’s recommendations. Primer sequences used were glyceraldehyde-3-phosphate dehydrogenase (*Gapdh*) (forward, 5′-ACTCCCACTCTTCCACCTTCG-3′, and reverse, 5′-ATGTAGGCCATGAGGTCCACC-3′), *Chac1* (forward, 5′-GTGGTGACCCTCCTTGAAGAC-3′, and reverse, 5′-AAGGTGACTTCCTTAGTGTCATAGC-3′), *Phgdh* (forward, 5′-TGGAGGAGATCTGGCCTCTC-3′, and reverse, 5′-GCCTCCTCGAGCACAGTTCA-3′), *Shmt2* (forward, 5′-CCCAAACTGGCCTCATCGAC-3′, and reverse, 5′-TGTGCCCTGACCTCATCACA-3′), *Shmt1* (forward, 5′-CCACCCTGTGGGCTTCTCAT-3′, and reverse, 5′-TTCTCCGAGGCAATCAGCTC-3′), *Atf4* (forward, 5′-CGGCTATGGATGATGGCTTG-3′, and reverse, 5′-TCAAGAGCTCATCTGGCATGG-3′), *Atf3* (forward, 5′-GCTGAGATTCGCCATCCAGA-3′, and reverse, 5′-TGTTTCGACACTTGGCAGCA-3′), *Ddit3* (forward, 5′-CCC TGCCTTTCACCTTGGAG-3′, and reverse, 5′-TGGCTCCT CTGTCAGCCAAG-3′), *Asns* (forward, 5′- GGAAGAACACAGACAGCGTGGTGA-3′, and reverse, 5′-TGGCGGCAGAGACAGGTAATAGGA-3′), *Eif4ebp1* (forward, 5′-GATGGAGTGTCGGAACTCACC-3′, and reverse, 5′-CATCTCAAATTGTGACTCTTC-3′), *Mthfd2* (forward, 5′-CTCCCAAAGAGCAGCTGAAG-3′, and reverse, 5′-AATTTGGGCTTTGCAGTGAC-3′), *Gadd45a* (forward, 5′-GGAGTCAGCGCACCATTACG-3′, and reverse, 5′-TTGATGTCGTTCTCGCAGCA-3′), *Cdkn1a* (forward, 5′-GTGGCCTTGTCGCT GTCTTG-3′, and reverse, 5′-AAGAGGCCTCCTGACCC ACA-3′). All reactions were performed in triplicate. Data analysis was carried out using the cycle threshold values of target gene expression normalized by *Gapdh* as the internal control. The expression levels of *Gapdh* mRNA were not significantly altered between Ser-supplemented and Ser-depleted Hep1-6 cells 6h (Appendix A) and 24 h (Appendix A) after the incubation when determined using *Atp5f1* as a reference gene.

### 2.3. siRNA Transfection

Hepa1-6 were transfected with Atf4 siRNA (30 nmol/L) (SASI_Mm02_00316863, Merck KGaA, Darmstadt, Germany) and Non-Targeting siRNA (scramble) (MISSION^®^ siRNA Universal Negative Control, Merck) in EMEM, using ScreenFect™siRNA (FUJIFILM Wako Chemicals, Ltd. Osaka, Japan) according to the manufacturer’s protocol. Cells were incubated with transfection reagent for 24 h before gene expression analysis.

### 2.4. Western Blot Analysis

Hepa1-6 were homogenized in a buffer containing 1.25 mM Tris-HCl (pH 7.6), 150 mM NaCl, 1% NP40, 1% sodium deoxycholate, 0.1% SDS, a protease inhibitor cocktail (Nacalai Tesque, Inc. Kyoto, Japan), and a phosphatase inhibitor cocktail (Nacalai Tesque, Inc., Kyoto, Japan). Homogenates were centrifuged at 20,000 g for 10 min to obtain total protein extracts, and concentrations were determined using a Protein Assay Bicinchoninate Kit (Nacalai Tesque, Inc., Kyoto, Japan). Protein samples were fractionated by 7.5% SDS-polyacrylamide gel electrophoresis and transferred onto a PVDF membrane (Bio-Rad, Hercules, CA, USA). To determine protein expression levels of Gapdh, Phgdh, cShmt, and mShmt, the blotted PVDF membrane was cut into several pieces at the corresponding molecular weight ranges, and then each membrane piece was probed with the following primary antibodies: anti-Phgdh (rabbit, 0.3 g/mL, provided by Dr. M. Watanabe at Hokkaido University) [11], anti-cShmt (also known as Shmt1; goat polyclonal, 1:500; Santa Cruz, TX, USA), anti-mShmt (also known as Shmt2; goat polyclonal, 1:1000; Santa Cruz, TX, USA), and anti-Gapdh (mouse monoclonal, Chemi-Con, 1:50000; Merck Millipore, Billerica, MA, USA). Bound antibodies were visualized with the Pierce SuperSignal West Pico Chemiluminescence Detection System (SuperSignal; Thermo Fisher Scientific, Inc., MA, USA) after incubating with the appropriate secondary antibodies conjugated with horseradish peroxidase (Cell Signaling Technology Japan K.K., Tokyo, Japan). When examined cShmt proteins, mShmt expression levels was detected first. Then, cShmt expression levels was probed after stripping the antibodies used for the detection of mShmt from the same PVDF membrane piece. The chemiluminescent signal was detected by exposure to X-ray films (FUJIFILM, Tokyo, Japan), and signal intensities were quantified with CS Image Analyzer 3 software (ATTO Corporation, Tokyo, Japan) as described previously [10].

### 2.5. Cell Growth Assay

Cell growth was determined as previously described [10]. Hepa1-6 cells were seeded at 1.0 × 10^4^ cells per well in a 24-well plate containing the complete medium and were incubated for 12 h. The medium was changed to Ser-supplemented or Ser-depleted Gly-supplemented condition for 24 h. Cell numbers were counted using the MTT assay solution (Cell counting kit-7, Dojindo Laboratories, Kumamoto, Japan) following the kit manual.

### 2.6. Amino Acid Analysis

Hepa1-6 were cultured in Ser-supplemented and -depleted condition for 6 h or 24 h, and washed with D-PBS, trypsinized, and centrifuged at 200 g for 5 min at 4 °C. The supernatants were removed, and the pellets were washed once with D-PBS. After a second centrifugation, the pellets were dissolved in 100 μL of Ultrapure Water (Wako Pure Chemical Industries Ltd., Osaka, Japan). To remove proteins in the sample, 50 μL of sample solution was added to 5 μL of 60% perchloric acid and centrifuged at 20,400 g for 20 min at 4 °C. After centrifugation, the supernatants were filtered through 0.20 μm filter unit (Merck Millipore, lerica, MA, USA) and were adjusted to pH 6–9 with 8 M potassium hydroxide, and then centrifuged at 20,400 g for 10 min at 4 °C.

To the 100 μL of cultured medium, 5 μL of 60% perchloric acid were added and centrifuged at 20,400 g for 20 min at 4 °C. After centrifugation, the supernatants were adjusted to pH 6–9 with 8 M potassium hydroxide, and then centrifuged at 20,400 g for 10 min at 4 °C. Subsequently, 10 μL sample solution was dried under reduced pressure. The dried residues were dissolved in 10 μL of 1 M sodium acetate-methanol-triethylamine (2:2:1), and dried under reduced pressure, dissolved in 20 μL of derivatization solution composed of methanol, distilled water-triethylamine-phenyl isothiocyanate (7:1:1:1), and incubated at room temperature for 20 min. The samples were dried again and dissolved in 200 μL of Pico-Tag Diluent (Nihon Waters K.K., Tokyo, Japan) and filtrated. The same methods were performed on standard solutions (type ANII, type B, L-asparagine, L-glutamine, and L-tryptophan; Wako Pure Chemical Industries Ltd., Osaka, Japan) which were diluted with Ultrapure Water. The samples were analyzed using HPLC system (two LC-10AD pumps (Shimadzu, Kyoto, Japan), CBM-20A system controller (Shimadzu, Kyoto, Japan), CTO-20A column oven (Shimadzu, Kyoto, Japan), pre-column (Wakopak Wakosil-PTC, 4 mm × 30 mm, Wako, Osaka, Japan), column (Wakopak Wakosil-PTC, 4 mm × 250 mm, Wako, Japan), manual injector (Rheodyne, Rohnert Park, CA, USA), SPD-10A UV-VIS detector (Shimadzu, Kyoto, Japan) and LCsolution software (Shimadzu, Kyoto, Japan) equipped with a PC). They were equilibrated with buffer A and eluted with a linear gradient of buffer B [water-acetonitrile-methanol (40:45:15)] (0, 3, 6, 9, and 100%; 0, 17, 28, 36 min respectively) at a flow rate of 1 mL/min at 46 °C. From 36 to 40 min, buffer B was used for washing. Then, it was equilibrate by buffer A for 20 min. The absorbance at 254 nm was measured.

### 2.7. Statistical Analyses

Differences between two groups were examined with the Student’s *t*-test. Differences among more than two groups were analyzed with a one-way analysis of variance followed by Dunnett’s post-hoc test and Tukey’s test. *p* values < 0.05 were considered significantly different. All statistics were performed using KaleidaGraph 4.0 (Synergy Software, Tokyo, Japan).

## 3. Results

We have recently demonstrated that Ser depletion led to upregulated transcription of *Atf4* and Atf4-target genes in KO-MEFs (Sayano et al., manuscript in preparation). We aimed to determine whether extracellular Ser depletion affects Atf4-target gene expression in other cell types expressing *Phgdh*. To see if Ser removal from extracellular environment alters the Atf4-mediated transcription, we measured Atf4-target genes *Chac1* mRNA by qRT-PCR in the mouse hepatoma cell line Hepa1-6 cells cultured under Ser-supplemented or -depleted conditions for 6 h. *Chac1* mRNA expression was significantly upregulated as Ser concentrations decreased in the culture media, when compared to Ser-supplemented conditions (Figure 1a). We measured the mRNA expression level of *Atf4* and its other target genes *Atf3*, *Ddit3* (Chop), *Eif4ebp1*, *Asns*, *Gadd45a*, *Mthfd2,* and *Cdkn1a*. The mRNA levels of *Atf4* and its targets were significantly elevated in the Hepa1-6 cells under Ser-depleted conditions (Figure 1b–i).

To test if Atf4 was involved in *Chac1* upregulation in Ser-depleted Hepa1-6, the cells were treated with siAtf4 to suppress Atf4, and the mRNA levels were measured 6 h after the depletion. Compared to scramble siRNA, siAtf4 treatment resulted in a 41% reduction of *Atf4* and a simultaneous 33% suppression of *Chac1* mRNA expression in Hepa1-6 under Ser-depleted condition (Figure 1j–k), suggesting that the *Chac1* mRNA induction is regulated primarily by Atf4 in response to Ser deficiency.

To clarify whether transcriptional activation regulates *Chac1* upregulation in Ser-depleted Hepa1-6, the cells were treated with actinomycin D (ActD) to inhibit mRNA synthesis. *Chac1* expression was markedly suppressed in Ser-depleted Hepa1-6 in the presence of ActD (Figure 2a), suggesting that the *Chac1* expression is a transcriptionally regulated in response to Ser deficiency. Notably, an upregulation of *Chac1* was obvious 2 h after the incubation with Ser-depleted medium, and then returned to basal level at 24 h (Figure 2b).

To confirm whether removing Ser specifically is responsible for these transcriptional changes, we examined the effect of replenishing Ser or other amino acids. Hepa1-6 cells were deprived of Ser for 6 h, and then switched to Ser-supplemented condition (400 μM). At subsequent time points (1, 2, 4 and 6 h) after Ser supplementation, *Chac1* mRNA levels were measured, and were significantly suppressed 2 h after Ser supplementation (Figure 3a). Further, the upregulation of *Chac1* in response to Ser depletion was not suppressed by the addition of Cys and Leu, but was remarkably suppressed by the addition Gly in a manner similar to Ser (Figure 3b). Because Ser depletion induces cell proliferation arrest in KO-MEFs [7], we measured the number of Hepa1-6 cells maintained under Ser- or Gly-supplemented and Ser-depleted conditions for 48 h. The cells grew more rapidly under both Ser and Gly supplemented condition than those under Ser depleted condition (Figure 3c).

We then analyzed intracellular and extracellular amino acid concentrations of Hepa1-6 cells under Ser-supplemented or -depleted condition for 6 h and 24 h. Intracellular Ser was not decreased in Ser-depleted condition for 6 h, but increased up to 2-fold at 24 h (Figure 4a). In contrast, intracellular Gly was decreased in Ser-depleted condition for 6 h, but recovered at 24 h (Figure 4b). In parallel with these changes, the level of extracellular Gly was significantly decreased at 6 h and 24 h under Ser-depleted conditions (Figure 4c). These observations led us to determine whether the activation of endogenous Ser synthesis from Gly is mediated by Shmt or de novo under Ser depletion. The isozymes Shmt1 and 2 are localized in cytosol and mitochondria, respectively. We then quantified mRNA levels of *Shmt1*, *Shmt2,* and *Phgdh* in Hepa1-6. *Phgdh* and *Shmt1* expression was not induced 2 h and 6 h after Ser depletion (Figure 5a,b), whereas *Shmt2* expression was significantly elevated under Ser depletion (Figure 5c). Phgdh and Shmt2 (mShmt) protein expression was significantly increased in Hepa1-6 in Ser-depleted condition for 6 h, whereas the protein level of Shmt1 (cShmt) was not significantly affected under Ser-depleted conditions (Figure 5d).

## 4. Discussion

In the present study, we demonstrate the response of *Phgdh*-expressing Hepa1-6 cells to extracellular Ser depletion. The observations suggest that Hepa1-6 cells can sense extracellular Ser deprivation and respond by inducing the expression of *Chac1* and other Atf4-target genes. Moreover, *Chac1* upregulation and growth arrest caused by Ser depletion is brought back to normal by supplementation of both Ser and Gly, the latter of which is likely converted to Ser by Shmt2.

Chac1, a pro-apoptotic factor carrying the enzymatic activity of glutathione-specific γ-glutamyl cyclotransferase (EC 4.3.2.7), is induced via the Atf4-Atf3-Chop axis activated by ISR [12,13], and whose overexpression induces oxidative stress and apoptosis via degradation of GSH [14]. Deprivation of Leu, Met, and Cys activates ISR and transcription of Atf4 and Atf4-target genes [15,16,17]. We have observed an upregulation of *Chac1* in KO-MEFs in an Atf4-dependent manner after decreasing the extracellular Ser concentration to 20% (Sayano et al. manuscript in preparation). Unlike KO-MEFs, intracellular Ser concentration did not decrease in Hepa1-6 cells after Ser depletion for at least 6 h (Figure 4a), whereas *Chac1* were upregulated. These findings support an idea that *Phgdh*-expressing cells like Hepa1-6 can sense changes in intracellular Ser availability that are caused by a lack of extracellular Ser by yet unknown mechanisms, which activate an Atf4-mediated transcriptional program quickly and transiently, leading to the expression of *Chac1* and other Atf4 target genes.

It should be noted that adding both Ser and Gly to Ser-depleted medium suppressed the *Chac1* upregulation in Hepa1-6 (Figure 3b). In a previous study, we demonstrated that the transcription of Atf4 target genes was stimulated in KO-MEFs after Ser deprivation [6], whereas Gly limitation in MCF7 cells failed to activate *ATF3* and *CHOP* translation, both of which are downstream of ATF4 [16]. The growth rate of Hepa1-6 in Gly-supplemented condition was similar to that of cells grown in Ser-supplemented conditions (Figure 3c). Moreover, the concentration of Gly was significantly decreased in the cells cultured for 6 h in Ser-depleted condition (Figure 4b), which was accompanied by an increase in Shmt2 expression (Figure 5d). These observations suggest that *Chac1* expression is suppressed in Hepa1-6 cells by Ser that is synthesized from Gly by Shmt2.

Gly is actively takes up in liver from the blood and transported to cytosol through cell membranes via glycine transporter 1 or alanine-serine-cysteine-1 transporter [17,18] and converted to Ser by mitochondrial and cytosolic Shmt. It was shown that mShmt (SHMT2 protein) in liver cancer cells converts Gly to Ser and its knockdown led to a significant reduction of intracellular Ser in Huh-7 hepatoma cells [19]. The enzyme activity of Phgdh in the liver is substantially enhanced in rats after feeding them a low-protein diet [20]. The present study showed that intracellular Gly was markedly reduced in Ser-depleted condition for 6 h and 24 h (Figure 4c) and Shmt2 and mShmt expression level were increased (Figure 5c,d) in Ser-depleted condition in Hepa1-6. Given these previous results and the present observations, when extracellular Ser availability is limited, Hepa1-6 cells sense extracellular Ser deficiency and activate both *de novo* Ser synthesis and Gly to Ser conversion via Shmt2 to maintain intracellular Ser concentration.

Intracellular Ser and Gly metabolism is essential for survival and proliferation in cancer cells. Melanoma and triple-negative breast cancer upregulate activity of Ser synthesis pathways by expanding the copy number of *Phgdh* [21,22]. Phgdh-dependent Ser synthesis is shown to contribute to redox homeostasis [2,9], folate and nucleotide synthesis [23,24,25], mitochondrial metabolism [26,27,28], and lipid metabolism [6,26], and to tumor growth [29,30,31]. Gly, a dispensable amino acid like Ser, has multiple and indispensable roles in a variety of anabolic reactions such as synthesis of purine, heme, glutathione [32,33]. Since our observations demonstrated that intracellular Ser was maintained in Hepa1-6 cells by consuming Gly via mShmt when extracellular Ser was depleted, the present study sheds light on the metabolic process of how to maintain intracellular Ser homeostasis in hepatomas and probably in hepatocytes. Future studies are needed to define the molecular mechanisms underlying the pathway used to sense extracellular Ser deficiency and the link between Ser deficiency signaling and expression of various genes.

## 5. Conclusions

In this study, we demonstrate that extracellular L-Serine depletion led to a transient activation of the Atf4-mediated transcriptional program in Hepa1-6 cells expressing Phgdh, but a retrieval of intracellular L-Serine via autonomous Glycine consumption antagonized the program by 24 h after the depletion.

## Figures and Tables

**Figure 1 nutrients-12-03018-f001:**
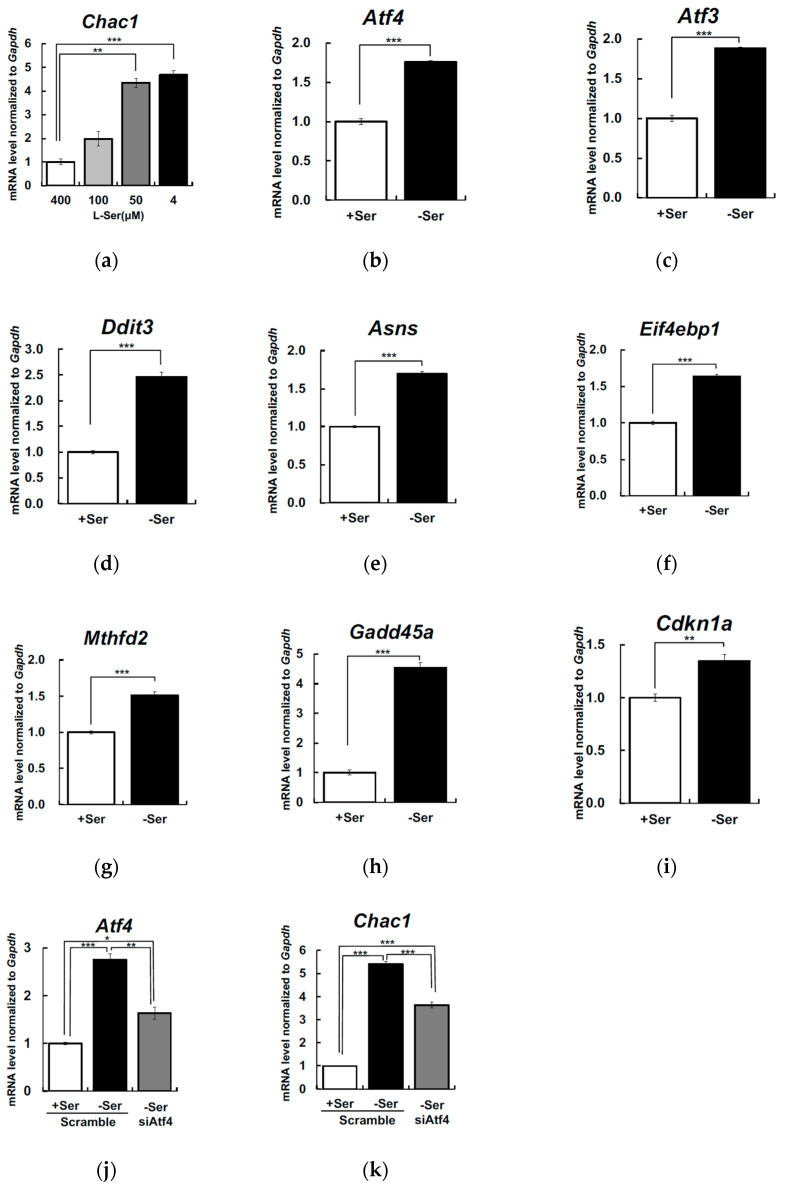
Extracellular L-Ser depletion induces cation transport regulator-like protein 1 (*Chac1*) and Atf4-target genes mRNA level in Hepa1-6.Hepa1-6 were cultured for 6 h in L-Ser-depleted (4 μM) or -supplemented (50, 100, 400 μM) Eagle’s Minimal Essential Medium (EMEM), and *Chac1* (**a**) mRNA levels were measured. Dunnett’s post hoc test, ** *p* < 0.005, *** *p* < 0.0005. Hepa1-6 were cultured under the L-Ser-depleted or -supplemented condition for 6 h, and *Atf4*, *Atf3*, *Ddit3* (**b**–**d**), and Atf4 targeted genes *Asns*, *Eif4ebp1*, *Mthfd2*, *Gadd45a* and *Cdkn1a* (**e**–**i**) mRNA were measured. Student’s *t*-test, ** *p* < 0.005, *** *p* < 0.0005. Hepa1-6 were cultured under the L-Ser-depleted or -supplemented condition for 6 h, and non-targeting siRNA and siAtf4 were transfected for 24 h, and *Atf4* and *Chac1* (**j**,**k**) mRNA were measured. Tukey’s test, * *p* < 0.05, ** *p* < 0.005, *** *p* < 0.0005.

**Figure 2 nutrients-12-03018-f002:**
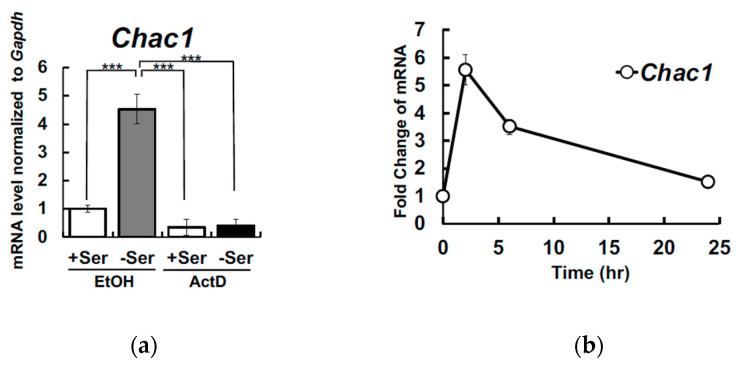
Extracellular L-Ser depletion transcriptionally and transiently induces *Chac1* mRNA expression in Hepa1-6. Hepa1-6 were cultured for 6 h under the L-Ser-depleted or -supplemented condition in the presence or absence of actinomycin D, and *Chac1* (**a**) mRNA levels were measured. Tukey’s test, *** *p* < 0.0005. Hepa1-6 were cultured for 2, 6 and 24 h under the L-Ser-depleted or -supplemented condition, and *Chac1* mRNA levels were measured (**b**).

**Figure 3 nutrients-12-03018-f003:**
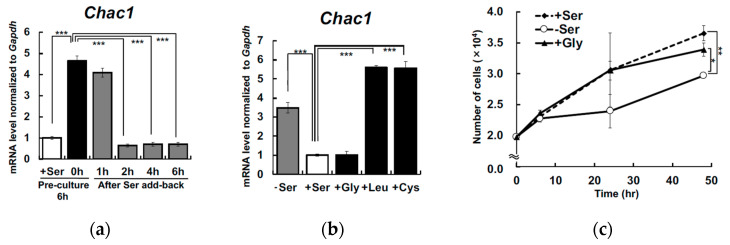
Addition of Ser and Gly suppress *Chac1* mRNA induction and cell growth arrest caused by extracellular L-Ser depletion in Hepa1-6. Hepa1-6 were cultured under the L-Ser-depleted or -supplemented condition for 6 h, and then L-Ser-depleted medium was replaced with L-Ser-supplemented medium, and *C**hac**1* (**a**) mRNA were measured after culturing for the indicated times. Hepa1-6 were cultured under the L-Ser-depleted or L-Ser-, Gly-, L-Leu-, or L-Cys-supplemented (final, 400 μM each) condition for 6 h, and *C**hac**1* (**b**) mRNA levels were measured. The number of live Hepa1-6 were counted after 24 h and 48 h incubation under the L-Ser-depleted or L-Ser- and Gly-supplemented (final, 400 μM each) condition (**c**). Dunnett’s post hoc test, * *p* < 0.05, ** *p* < 0.005, *** *p* < 0.0005.

**Figure 4 nutrients-12-03018-f004:**
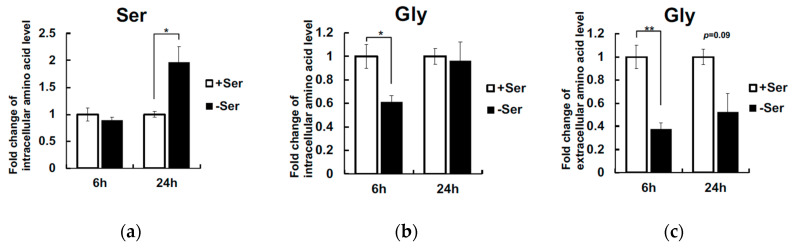
Extracellular L-Ser depletion alters the intracellular and extracellular Ser and Gly level in Hepa1-6. Hepa1-6 were cultured for 6 h and 24 h under the L-Ser-depleted or -supplemented condition, and concentration of intracellular L-Ser (**a**), intracellular Gly (**b**), and extracellular Gly (**c**) were measured. Student’s *t*-test, * *p* < 0.05, ** *p* < 0.005.

**Figure 5 nutrients-12-03018-f005:**
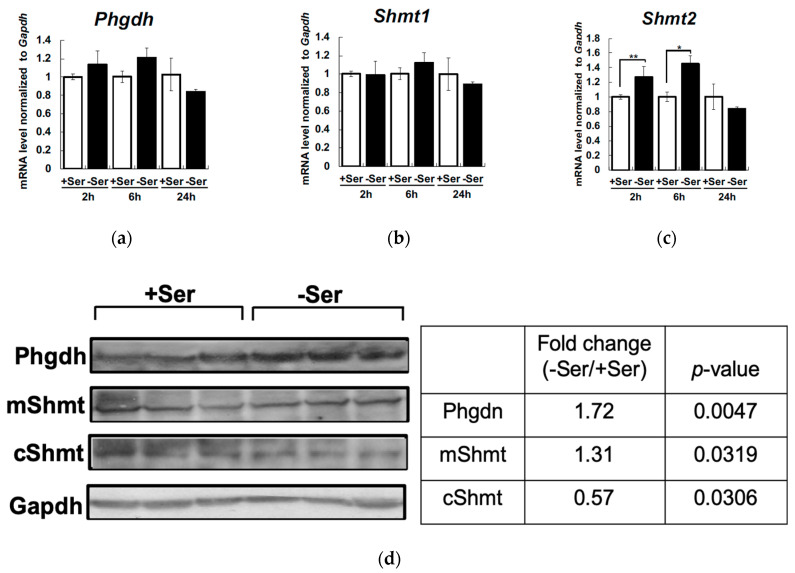
Extracellular L-Ser depletion activate endogenous Ser synthesis in Hepa1-6. Hepa1-6 were cultured under the L-Ser-depleted or -supplemented condition for 6 h, and D-3-phosphoglycerate dehydrogenase (*Phgdh)* (**a**), serine hydroxymethyltransferase 1 (*Shmt1)* (**b**), and *Shmt2* (**c**) mRNA levels were measured. Hepa1-6 were cultured for 6 h under the L-Ser-depleted or -supplemented condition, and the protein levels of Phgdh, Shmt1, and Shmt2 were determined after normalization to the amount of Gapdh (**d**). Student’s *t*-test.

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
