# Peer review of "Transcriptional Activation of Chac1 and Other Atf4-Target Genes Induced by Extracellular l-Serine Depletion is negated with Glycine Consumption in Hepa1-6 Hepatocarcinoma Cells"

_nutrients, 2020, doi:10.3390/nu12103018_

Round 1

Reviewer 1 Report

At the bottom of the document I include my previous questions and the answers of the authors of the manuscript (After the dotted line) .

In my opinion, the changes made by the authors respond to the questions raised in the previous review.

It would only include a change prior to final acceptance. In figure five, I would build a panel with the three proteins (phgdh, cShmt and mShmt). I would include the GAPDH at the bottom of the panel. And on one side of the panel the quantification.

I believe that this work is suitable for publication in the journal.

Best regards.

---------------------------------------------------------------------------------------------------------------------------------------------------
The introduction, materials and methods in the paper work very well, especially the part that correspond to the amino acid analysis.

Response: Thank you for your positive comments about the amino acid analysis method.

2. Results are good, and the figures are clear but the results got in the experiments should be described objectively, not conclusions were made; In addition, the part of discussion is not well discussed combined with results and references and should make some modifications. I consider it necessary to include references to the results and figures in the discussion section.

Response: We agree with your comment. We have added a speculation of a possible mechanism of Chac1 induction caused by Ser deficiency by citing some references in the second and third paragraphs of the Discussion (page 10 and 11). In the 4th paragraph of the Discussion, to link the present results with reported observations of Ser and Gly metabolism in the liver, we have added the following sentences: Page 11, lane 289 “The present study showed that intracellular Gly was markedly reduced in Ser-depleted condition for 6 h and 24h (Fig.4C) and Shmt2 and mShmt expression level were increased (Fig.5C and Fig.5F) in Ser-depleted condition in Hepa1-6. Given these previous results and the present observations, when extracellular Ser availability is limited, Hepa1-6 cells sense extracellular Ser deficiency and activate both de novo Ser synthesis and Gly to Ser conversion via Shmt2 to maintain intracellular Ser concentration.”

Page 12, lane 302 “Since our observations demonstrated that intracellular Ser was maintained in Hepa1-6 cells by consuming Gly via mShmt when extracellular Ser was depleted, the present study sheds light on the metabolic process of how to maintain intracellular Ser homeostasis in hepatomas and probably in hepatocytes.”.

3. Line 90 of the document, in the Materials and Methods (WB) section, does not indicate the membrane image acquisition system. Is the traditional system (film)?; is a digital image acquisition system?. I also indicate some commentary on Figure 5F, which also refers to protein analysis (WB). These panels do not refer to the protein quantification system. The system of quantification and standardization versus gapdh is also not referred to. In my opinion should be reconstructed the figure including the three membranes (phgdh, cShmt and mShmt) and only one GAPDH (it is the same in the three membranes). Quantification (next to it) can and should be included. Finally, I would recommend that you do not collapse the images. the impression you get when you view them is that they are narrow.

Response: We have acquired the membrane image by using the traditional system (film). Protein expression level of GAPDH, Phgdh, cShmt and mShmt were obtained by cutting the membrane into small peaces at the objective molecular weight from a single membrane. The original manuscript was inadequate in describing this point.

The following sentence has been added to describe the method of western blot in more detail in the revised manuscript: Page 6 lane144 To determine protein expression levels of Gapdh, Phgdh, cShmt and mShmt, the blotted PVDF membrane was cut into several pieces at the corresponding molecular weight ranges, and then each membrane piece was probed with the following primary antibodies:

4. Remove spaces in the text. Line 250: REGENDS must be replaced with LEGENDS.

Response: We have fixed this type error in page 18 lane 459 of the revised manuscript.

5. This section of the manuscript refers to a manuscript in preparation. Reference is made to results that have not been revised or published. Response: We need to refer to this unpublished data in this manuscript because it is essential reference to the planning of this study. These unpublished data will be published in elsewhere soon.

6. The language is not fluent, suggesting that the paper should be language-edited by native english-speaker editor or colleagues. Please make large revisions, especially in the parts of results and discussion. After making large revisions, the paper may be considered for publication Response: This manuscript was properly edited at the time of submission by an editor in Editage, a company that specializes in proofreading scientific papers.

Author Response

It would only include a change prior to final acceptance. In figure five, I would build a panel with the three proteins (phgdh, cShmt and mShmt). I would include the GAPDH at the bottom of the panel. And on one side of the panel the quantification.

Response:

According to the reviewer’s suggestion, we have revised Figure 5D, E and F.

We have aggregated the image data of Phgdh, cShmt,  mShmt and Gapdh blots into a single column, with quantitative normlized values of -Ser/+Ser and their p-values on the right side.

Reviewer 2 Report

The authors have addressed all my previous concerns.

Author Response

Thank you for confirming our response to your comments,

We have slightly revised the manuscript in Figure 5.

We have continued to incorporate your comments into our new manuscript.

Reviewer 3 Report

All the points raised were properly addressed.

Author Response

Thank you for confirming our response to your comments,

We have slightly revised the manuscript in Figure 5.

We have continued to incorporate your comments into our new manuscript.

This manuscript is a resubmission of an earlier submission. The following is a list of the peer review reports and author responses from that submission.

Round 1

Reviewer 1 Report

Nutrients (ISSN 2072-6643)

Manuscript ID; nutrients-837032

Transcriptional activation of Fgf21 and other Atf4-target genes induced by extracellular L-serine depletion is negated with glycine consumption in Hepa1-6 hepatocarcinoma cells.

Momoko Hamano * , Shozo Tomonaga , Hiroaki Oda , Hisanori Kato , Shigeki Furuya

Mouse embryonic fibroblasts lacking D-3-phosphoglycerate dehydrogenase (Phgdh), which catalyzes the first step of de novo synthesis of L-serine, are particularly sensitive to depletion of extracellular L-serine. In these cells, depletion of L-serine leads to a rapid reduction of intracellular L-serine, cell growth arrest, and altered expression of a wide variety of genes. However, it remains unclear whether reduced availability of extracellular L-serine elicits such responses in other cell types expressing Phgdh. Here, we show in the mouse hepatoma cell line Hepa1-6 that extracellular L-serine depletion transiently induced transcriptional activation of Atf4-target genes, including Fgf21 and Chac1. Expression levels of these genes returned to normal 24 hours after L-serine depletion, and were suppressed by the addition of L-serine or glycine in the medium. Extracellular L-serine depletion caused a reduction of extracellular and intracellular glycine levels but maintained intracellular L-serine levels in the cells. Further, Phgdh and Shmt2 were upregulated after L-serine depletion. These results led us to conclude that the Atf4-mediated gene expression program is activated by extracellular L-serine depletion in Hepa1-6 cells expressing Phgdh, but is antagonized by the subsequent upregulation of L-serine synthesis, mainly from autonomous glycine consumption.

It is a topic of interest to the researchers in the related area but the paper needs minor improvements before acceptance for publication. My detailed comments are as follows:

  1. The introduction, materials and methods in the paper work very well, especially the part that correspond to the amino acid analysis.
  2. Results are good, and the figures are clear but the results got in the experiments should be described objectively, not conclusions were made; In addition, the part of dicussion is not well discussed combined with results and references and should make some modifications. I consider it necessary to include references to the results and figures in the discussion section.
  3. Line 90 of the document, in the Materials and Methods (WB) section, does not indicate the membrane image acquisition system. Is the traditional system (film)?; is a digital image acquisition system?. I also indicate some commentary on Figure 5F, which also refers to protein analysis (WB). These panels do not refer to the protein quantification system. The system of quantification and standardization versus gapdh is also not referred to. In my opinion should be reconstructed the figure including the three membranes (phgdh, cShmt and mShmt) and only one GAPDH (it is the same in the three membranes). Quantification (next to it) can and should be included. Finally, I would recommend that you do not collapse the images. the impression you get when you view them is that they are narrow.
  4. Remove spaces in the text. Line 250: REGENDS must be replaced with LEGENDS.
  5. This section of the manuscript refers to a manuscript in preparation. Reference is made to results that have not been revised or published.
  6. The language is not fluent, suggesting that the paper should be language-edited by native english-speaker editor or colleagues. Please make large revisions, especially in the parts of results and discussion. After making large revisions, the paper may be considered for publication

Reviewer 2 Report

The study by Hamano et al reports the transcriptional response of a few selected genes to depletion of extracellular L-Serine in a mouse hepatoma cell line.

Major concerns

  1. The introduction is missing recent relevant studies showing the pleiotropic function of serine glycine one-carbon metabolism in cancer. The authors cite a review from 2014 and two studies from 2013 and 2016, respectively. I would suggest citing one or two more current reviews in the field that summarize the current views, which extends well beyond nucleotide, ATP, and glutathione production.
  2. Gapdh is a metabolic enzyme and might not be a good housekeeping gene under the conditions tested in this study (PMC1937003). The authors should provide evidence that Gapdh as a housekeeping gene for the conditions included in this study.
  3. The authors do not explicitly show in this study that Fgf21 and Chac1 expression depend on ATF4 activity upon L-Serine depletion. Measuring the levels of Fgf21 and Chac1 in ATF4 deficient cells +/- L-Serine depletion would be required to support the claims stated in the manuscript.
  4. In Figure 2A: Fgf21 mRNA in +Ser vs – Ser conditions in Actinomycin D treated samples shows an effect. These data would argue that the upregulation of this transcript does not entirely rely on transcriptional activation. It has been proposed that mRNAs of the serine biosynthesis pathway rely on mechanisms of mRNA stability (https://www.nature.com/articles/s41422-019-0257-1). The authors need to address this concern.
  5. Figure 3C: It is not clear to this reviewer how Glycine add-back (400uM) is able to suppress Fgf21 and Chac1 upregulation if glycine was never removed from the media (Glycine concentration in EMEM media is 100uM - https://www.atcc.org/~/media/28398321D72C40969560D673E9EF94C3.ashx). Authors should repeat the experiment in -Ser-Gly depleted conditions and add back Serine and Glycine individually and combined, to physiologically relevant concentrations.
  6. The authors use Hepa1-6 cells from mouse. It is well known that mouse contain a functional TDH enzyme that is able to replenish Glycine/Serine pools from Threonine, but the function of this enzyme is not conserved in humans. Authors should include data recapitulating key findings on glycine and serine deprivation using a human hepatoma cell line.

Minor concerns

  1. The title has been cut, correct.
  2. How much ATF4 and PHGDH do Hepa1-6 express compared to MEFs at the protein and mRNA levels in normal growth conditions?
  3. Do Hepa1-6 increase glucose consumption in response to serine deprivation?
  4. It is not clear how many times each experiment has been repeated, and whether the error bars represent technical or biological variability.
  5. Related to Figure 3C: Does Glycine rescue Fgf21 and Chac1 expression in cells without SHMT1/2?

Reviewer 3 Report

The manuscript by Hamano and colleagues demonstrated the impact of extracellular serine on the transcriptional activation of ATF4-target genes. The authors cultured hepatoma cells in the presence and absence of serine and showed that Atf4-target genes as well as Phgdh and Shmt2 were upregulated in the absence of serine. The authors conclude that medium serine impacts ATF4-mediated gene expression and intracellular glycine consumption. The findings may provide useful information on amino acid availability and ATF4 regulation in the field of cancer.

Serine/glycine metabolism is of high interest to the research community. However, the current manuscript lacks essential new findings, and the current data do not support the main conclusions. Some observations have been published by the same group in an earlier publication using a different (Phgdh expressing) cell model (Sayano et al 2013). In their current manuscript, the withdrawal of extracellular serine does not affect intracellular serine levels but induces ATF4-related genes. However, the authors have linked decreased intracellular serine levels to ATF4 activation in an unpublished manuscript (Sayano et al). Further, the addition of glycine rescues the phenotype (Fig 3D) and, thus, the impact of serine on ATF4-target genes is not clear. Additional experiments and discussions are required to clarify the time-dependent effect of serine withdrawal on metabolism and gene/protein expression.

  1. The authors refer to unpublished work in the result and discussion part. As such, I cannot comment on those statements.
  2. Expression levels and growth were not significantly affected in -Ser conditions at 24h (Fig 2C, 3E). Thus, it is unclear how changes in gene expression (at 6h) may be linked to the growth defect. Does serine addition rescue growth when adding back at 6h?
  3. The current manuscript does not provide sufficient data to support “glycine consumption”. The authors have measured glycine levels in medium lacking serine and glycine, indicating that cells secrete glycine (in addition to conversion to serine). Flux studies are needed as serine may be derived from other sources.
  4. Gene expression of Phgdh was not affected after 6h in cultures without serine. However, protein levels were significantly increased (Fig 5D). How does serine withdrawal affect protein levels of ATF4-targets? Does the addition of glycine have similar effects on protein levels (Figure 5)?
  5. The authors may want to include additional details on experimental design to clarify if experiments were performed in the absence or presence of glycine. For instance, it is not clear if serine was present in condition treated with glycine (Fig 3C). Did the author mean “serine or glycine” instead of “serine and glycine” (line 210)?

Minor points:

  1. Crosslink to some figures is missing. For instance, the manuscript does not refer to Figure panel 1D.

Reviewer 4 Report

In this manuscript Dr. Hamano and collaborators evaluate the Atf4-mediated transcriptional response to L-Serine depletion in the mouse hepatoma cell line Hepa1-6. The authors also evaluated the changes in cell proliferation, intra- and extra-cellular amino acid concentration, and changes in protein levels of key enzymes involved in the synthesis of Serine and Glycine.

Overall, the manuscript  includes a good experimental design and well performed experiments. However, some points must be addressed:

  • The author must include the primers used for all the targets evaluated. The primers sequences for Ddit3, Eif4ebp1, Asns, Gadd45a, Mthfd2, and Cdkn1a were no provided. In addition, if possible, melt curves should be included to demonstrate target.
  • In the result section the authors state: ‘whereas the protein level of Shmt1 187 (cShmt) was not significantly affected under Ser-depleted conditions (Fig. 5F)’. However, western blot analysis for cShmt correspond to Figure 5E in which, in turn, a significant decrease is depicted in the Serine depletion group.
  • In figures 5D,E and F. Images of Western blot results are presented, showing just two replicate per condition. Given the variability inherent to the technique more replicate should be included to obtain valid estimations. In addition, it can be seen that the exact same image for housekeeping blot loading control (GAPDH) was used for analysis of Phgdh, cShmt and mShmt). Given that the cShmt and mShmt have the same molecular weight size and both antibodies were produced in goat, it seems impossible to detect both proteins in the same membrane blot. How can the authors used the exact same loading control image?. A picture of the entire membrane picture and/or film should be included. If membrane stripping was used, also both membrane pictures and/or films should be included, showing the differences expected in GAPDH images.
  • qRT-PCR for Asns, Eif4ebp1, Mthfd2, Gadd45a, and Cdkn1a are described in the text to correspond to Figure 1C, however they are labeled as Figure 1D.